# Public misconceptions about dyslexia: The role of intuitive psychology

**Iris Berent** *, **Melanie Platt**

Department of Psychology, Northeastern University, Boston, Massachusetts, United States of America

* i.berent@northeastern.edu

## Abstract

Despite advances in its scientific understanding, dyslexia is still associated with rampant public misconceptions. Here, we trace these misconceptions to the interaction between two intuitive psychological principles: Dualism and Essentialism. We hypothesize that people essentialize dyslexia symptoms that they anchor in the body. Experiment 1 shows that, when dyslexia is associated with visual confusions (*b/d* reversals)—symptoms that are naturally viewed as embodied (in the eyes), laypeople consider dyslexia as more severe, immutable, biological, and heritable, compared to when dyslexia is linked to difficulties with phonological decoding (a symptom seen as less strongly embodied). Experiments 2–3 show that the embodiment of symptoms plays a causal role in promoting essentialist thinking. Experiment 2 shows that, when participants are provided evidence that the symptoms of dyslexia are embodied (i.e., they "show up" in a brain scan), people are more likely to consider dyslexia as heritable compared to when the same symptoms are diagnosed behaviorally (without any explicit evidence for the body). Finally, Experiment 3 shows that reasoning about the severity of dyslexia symptoms can be modulated by manipulating people's attitudes about the mind/body links, generally. These results show how public attitudes towards psychological disorders arise from the very principles that make the mind tick.

## Introduction

Dyslexia is a common reading disorder affecting approximately 7% of the population (for reviews: [1, 2]). Advances in reading science have shed light on the symptoms of dyslexia and their origin. Remarkably, the public understanding of dyslexia differs markedly and from the evidence emerging from reading research [3–5].

These public misconceptions of dyslexia raise a number of questions—both translational and foundational. At the translational level, the misconceptions about dyslexia present a worrisome obstacle for treatment. Although dyslexia is a heritable disorder, reading research has made it clear that its symptoms can be remedied by appropriate interventions [6]. But to ensure that children with dyslexia promptly obtain appropriate interventions, it is critical that the general public—parents, educators, and legislators—are aware of dyslexia and its symptoms. Public misconceptions of dyslexia can prevent affected individuals from obtaining appropriate interventions.

**Data Availability Statement:** Data are attached as SM.

**Funding:** The author(s) received no specific funding for this work.

**Competing interests:** The authors have declared that no competing interests exist.

The misconceptions of dyslexia are also of inherent interest to cognitive science. As we next show, laypeople's attitudes towards dyslexia do not arise only from innocent ignorance about the science of reading. Rather, these misconceptions are guided by systematic, but faulty intuitive assumptions about how the mind works, generally. In fact, the principles we invoke to explain the misconceptions of dyslexia can also explain public misconceptions of major psychiatric disorders. As such, laypeople's attitudes towards dyslexia can shed light on how laypeople reason about their own psyche in health and disease.

In what follows, we first review the conclusions emerging from reading science about dyslexia, and contrast them with laypeople's views. To explain the attitudes towards dyslexia, we next briefly summarize the findings from two related literatures—one on laypeople's misconceptions about the workings of typical minds, and another, about psychiatric disorders. Against this backdrop, we propose a novel account of how the misconceptions towards dyslexia arise from intuitive psychology. We finally test this proposal in a series of experiments.

## Dyslexia and its origins: Scientific facts and public fiction

To clarify the nature of dyslexia, it is first necessary to briefly consider typical reading. Reading science makes it clear that when skilled readers identify printed English words, they decode their phonological structure from print by mapping letters into phonemes (e.g., linking the *c* in *cat* to the phoneme /k/, in *kick*; [7, 8]).

In people with dyslexia, phonological decoding is often disrupted [1, 9]. This difficulty is especially noticeable in reading novel words like *blin*; since these words are unfamiliar, their pronunciation cannot be retrieved from memory—it can only be obtained by phonological decoding, "from scratch". And since people with dyslexia struggle with phonological decoding, they find novel words particularly challenging [10].

Dyslexia, to be clear, compromises not only reading but also speech perception [11]. Other research has documented difficulties in visual perception [12, 13], phonemic awareness [7], attention and working memory [14], morphological processing [15], and text comprehension [16]. Nonetheless, in orthographies like English, where the mapping from letters to sounds is not fully predictable, phonological decoding from print is, by far, the most common symptom of dyslexia [1, 9].

If laypeople were aware of this scientific fact, then they should have considered phonological difficulties as the most characteristic challenge in dyslexia. As noted, for English, dyslexia typically compromises phonological decoding, and most studies of laypeople's attitudes towards dyslexia have been conducted among English speakers. But in reality, many laypeople believe that dyslexia is a form of "word blindness" [4] that results from "troubles with vision" [5].

In a large US study, most participants—laypeople and educators—believed that a "common sign of dyslexia is seeing letters backwards" [17]. Similarly, British student teachers stated that "colored overlays and/or tinted glasses were helpful to individuals with dyslexia", and that "eye tracking exercises are effective in remediating dyslexia-caused difficulties" [3].

People are likewise confused about the *etiology* of the disorder. Reading science shows that dyslexia is a hereditary disorder that is associated with a number of candidate genes [18, 19]. To be clear, reading is a learned skill—no infant is born knowing that "b" sounds like /b/. But since reading recruits the speech and language brain network, the mechanisms that supports reading and phonological decoding are heritable [19, 20].

Laypeople correctly recognize that "people cannot help being dyslexic—it is in their genetic make-up" [4]. But upon closer inspection, it is evident that their understanding of these notions is fragile. For example, student teachers were not quite sure whether "dyslexic parents

are more likely than non-dyslexic parents to have children with dyslexia" [3] and they tended to disagree with the statement that "dyslexia is caused by inherited, faulty genes with evidence coming from studies of twins" [4]. Most surprisingly, laypeople do not necessarily view dyslexia as a brain disorder [4, 5]. For example, participants were uncertain that the disorder is caused by "medical neurological factors" [5] and that "brain scan studies show that dyslexics' brains work differently from those of non-dyslexics" [4] xe.

These observations raise two questions. First, why do such misconceptions arise? Second, are laypeople's attitudes about the symptoms of dyslexia (as either visual- or decoding difficulties) linked to their views about its origins (innate or acquired) and its manifestation in the brain? For example, are people more likely to view dyslexia as a heritable brain disorder if they consider it primarily a visual disorder (as opposed to a disorder affecting phonological decoding)?

## Laypeople's misconceptions about the mind in health and disease

A priori, it is conceivable that laypeople's attitudes towards the symptoms of dyslexia could arise from multiple sources, including their own experiences with learning to read, discussions in the media, and science. Past research, however, has shown that reasoning about the mind is shaped by intuitive psychology [21–25], and these intuitive principles can further guide reasoning about another class of disorders—psychiatric disorders.

These beliefs arise from the tension between two fundamental principles of intuitive psychology: Dualism and Essentialism [21, 26]. To be clear, Dualism and Essentialism are ***psychological*** principles, distinct from the philosophical doctrines by the same name. And indeed, these principles are utterly ***tacit***—people are largely unaware of these biases, and indeed, they are evident in young children, and possibly infants [25, 27, 28]. Nonetheless, intuitive Dualism and Essentialism demonstrably guide laypeople's reasoning about mind, body, and inheritance.

In what follows, we review the role of Dualism and Essentialism in reasoning about typical psychological traits and psychiatric disorders. We next move to consider their potential role in the case of dyslexia.

**Intuitive reasoning about typical psychological traits.** *a. The role of Dualism.* Past research suggests that people are intuitive ***Dualist***s [24] inasmuch as they associate physical traits with the material body, whereas they tend to link psychological traits to the ethereal mind.

To shed light on Dualism, past research has invited people to imagine a situation that would replicate a layperson's body (following [29])—of interest is which of the person's features would transfer to the replica. If people believe that the person's traits all form part of the person's body, then they ought to conclude that these traits should all transfer to the replica. Results, however, show that people do not consider all traits as embodied. They believe that the replica will preserve physical traits, such as hair color, but not psychological traits, such as the memory of childhood friend [30, 31].

To demonstrate that these intuitions do not arise simply because people are somehow confused by psychological traits, in another set of studies, people were invited to reason about the outcomes of scenarios that manipulate the mind—either a "mind switching" scenario [32, 33] or the afterlife [34]. If people believe that psychological traits reside in the mind (and not in the body), then once the manipulation targets the mind, it is psychological traits that ought to be most likely to persist. This, indeed, is precisely what the results show [32–34]. The ***double dissociation*** between psychological and bodily traits in scenarios that manipulate the mind

and those that manipulate the body suggests that people indeed consider psychological traits as ethereal, distinct from the body.

Interestingly, psychological traits are not all viewed as equally embodied. Critical to this intuitive view is the distinction between two sets of traits: one set of traits are epistemic—those that capture what a person knows or believes; another set of traits are sensory (e.g., audition, vision). Results show that people anchor sensory traits in the body, but they consider epistemic traits as ethereal, distinct from the body. For example, laypeople believe that sensory traits (e.g., audition, vision) are more likely to reside in the brain and to transfer to a replica of a person's body than epistemic traits (e.g., having concepts such as "a person", "number", [35]). But when people reason about the afterlife—a situation that selectively targets the mind, they now consider epistemic traits as the ones that are more likely to persist than sensory ones [34, 35]. Altogether, these findings demonstrate that *epistemic traits are considered ethereal compared to sensory traits*.

*b. The role of Essentialism*. The Dualist beliefs about the anchoring of the psyche in the body can further shape laypeople's attitudes towards the innate origins of the psyche and its manifestation in disease. These innateness beliefs arise from the interaction of Dualism with a second intuitive principle—***Essentialism***.

A large literature shows that people attribute inheritance to some immutable essence that offspring obtain from their biological parents (e.g., [25, 36–38]). Young children believe that a pig is more likely to exhibit the physical properties of its biological rather than adopting parents, and they appear to attribute these properties to some immutable essence that lies in the animal's insides [38].

People, however, seem to believe that this immutable essence lies not in the mind but in the body (for review, [39]). Indeed, even infants believe that agents must possess some physical insides (i.e., they cannot be hollow; [27]). Likewise, young children believe that the essence of an animal is localized at its center [40], and in some cultures, this essence is associated with some specific substance (e.g., blood; [41]). "Insides", center, and bodily substances all define the body. Together, these results suggest that *per Essentialism, inherited biological traits reside in the body*.

*c. A perfect storm*. The possibility that Dualism and Essentialism can each constrain intuitive psychology is well known—two large literatures have explored these two possibilities. What has now been previous noticed, however, is that these beliefs about body and mind, on the one hand, and essence, on the other, *interact*. A new proposal examines this perfect storm [21, 26].

Recall that, per Dualism, epistemic traits are considered ethereal, distinct from the body. Essentialism, however, demands that innate traits be anchored in the body. Combining the two, it thus follows that epistemic traits cannot be innate (see Fig 1).

This prediction is borne out by the outcomes of a series of studies [35, 42–45]. For example, when people seek to determine which psychological traits are likely innate, they consider epistemic traits (e.g., a notion of a person) as less likely to be innate than sensory and motor capacities [35, 46, 47]. Moreover, attitudes concerning the innateness of traits and their bodily manifestations are tightly linked—the more likely the trait is to be viewed as materially instantiated in the body, the more likely it is to be considered innate [35, 42].

The tension between Dualism and Essentialism not only captures laypeople's attitudes towards typical psychological traits but also towards atypical ones—those seen in disorders. We first briefly review the implications to psychiatric disorders; we next move to consider dyslexia.

**Dualism and Essentialism bias reasoning about psychiatric disorders.** A large literature has explored the role of Dualism and Essentialism in laypeople's reasoning about mental

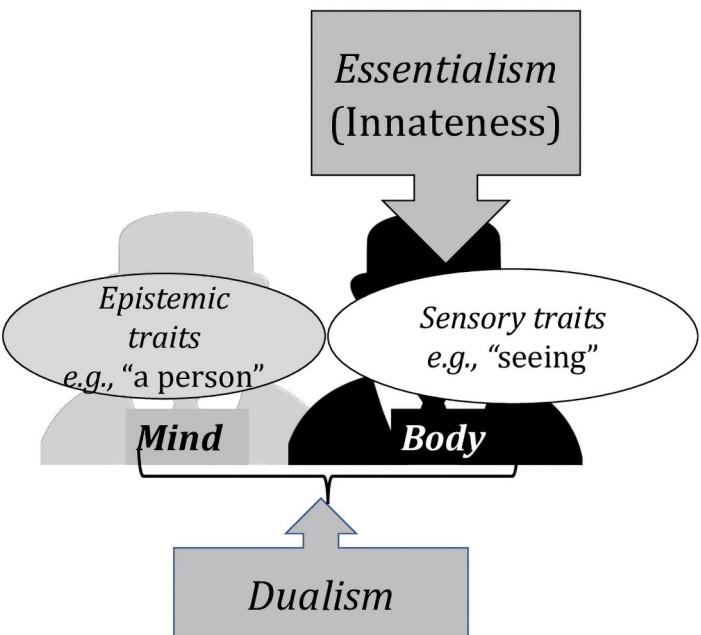

**Fig 1. The role of Dualism and Essentialism in reasoning about typical psychological trait.**

disorders. Dualism has been invoked to explain the curious finding that, when psychiatric symptoms are linked to a biological source, laypeople's attitudes towards the symptoms and patients are generally more *negative*: people consider biological symptoms as less controllable by the patient [48, 49] and as less likely to benefit from psychotherapy [48, 50, 51].

Essentialism, in turn, has been blamed for laypeople's belief that biogenetic psychiatric disorders are immutable—biogenetic symptoms are considered lengthier [51, 52], less responsive to treatment [53, 54], and more characteristic of patients' biological families [55] And since Essentialism would suggest that the patient's core is different from one's own, Essentialism can further promote social stigma [54, 56].

Critically, there is evidence that these essentialist beliefs do not arise simply because biogenetic explanations offer evidence that the disorder in question is genetically based. Indeed, biological explanations can promote negative attitudes even when they merely link the disorder to the brain (without suggesting any explicit genetic origin; [54]). To evaluate this possibility, in a recent set of studies, we invited participants to reason about psychiatric disorders [44]. In one condition, the diagnosis was informed by a brain test (i.e., a test that explicitly references the body); in another, it was informed by a matched behavioral test (a test that does not reference the body explicitly). Although the tests were fully matched—all they suggested is whether or not the person had a disorder, people were more likely to exhibit negative attitudes towards patients, and to view the disorder as heritable and lengthier (i.e., immutability; [44]).

Immutability and innateness are, of course, the telltale signs of essentialism. Interestingly, this **essentialist** thinking is triggered by the belief that the disorder in question resides **in the body** (as suggested by a brain test). Since people are less likely to exhibit essentialist thinking for symptoms that are diagnosed by a behavioral test (when the test does not invoke the body, explicitly), these results suggest that people believe some psychological traits exist only "in the mind". *Essentialist thinking, then, interacts with Dualism.*

In light of the role of Dualism and Essentialism in laypeople's reasoning about the psyche—about typical psychological traits and major psychiatric disorders—here, we ask whether these same principles could further explain laypeople's puzzling attitudes towards dyslexia.

**The role of Dualism and Essentialism in dyslexia.** The possibility that intuitive psychology could bias laypeople's attitudes towards dyslexia is not entirely new. Past research has invoked Essentialism to explain why laypeople consider dyslexia as immutable when its symptoms are linked to biological causes [57]. While this proposal can certainly explain why biological symptoms are considered immutable, it leaves two major questions unanswered. First, why do people wrongly assume that dyslexia results from *visual* confusions (rather than from troubles with phonological decoding, [3–5, 17])? Second, why do people sometimes struggle to link dyslexia to the *brain* [4, 5]?.

The interaction of Essentialism with Dualism can account for both facts. Recall that, in intuitive psychology, some psychological traits are considered more embodied than others—sensory traits, in particular, are considered more embodied than epistemic traits. All this leads us to predict that, when the Dualist considers how reading works, they should *align sensory traits like "vision" with the body, but they should link epistemic traits like "knowing the correspondence between letters and sounds" with the ethereal mind.*

If people further assume (correctly) that dyslexia is a heritable disorder that runs in families, and if (per Essentialism), they require innate traits to be embodied, then people should be more likely to link dyslexia to (embodied) troubles with "vision" compared to (ethereal) troubles with linking letters and sounds. And, since per Essentialism, what's innate is further immutable, this same analysis further predicts that *the visual symptoms of dyslexia should be considered as more severe and immutable than troubles with phonological decoding* (Fig 2).

This same line of reasoning can also explain why people have troubles linking dyslexia to the brain [4, 5] and predicts that this difficulty should be particularly pronounced when people link dyslexia to phonological decoding. In laypeople's view, phonological decoding is entirely learned from experience—it invokes no innate learning mechanisms (contrary to the evidence from science). And since people associate innateness with the body, they might conclude that what is learned cannot form part of the body. Since phonological decoding is considered

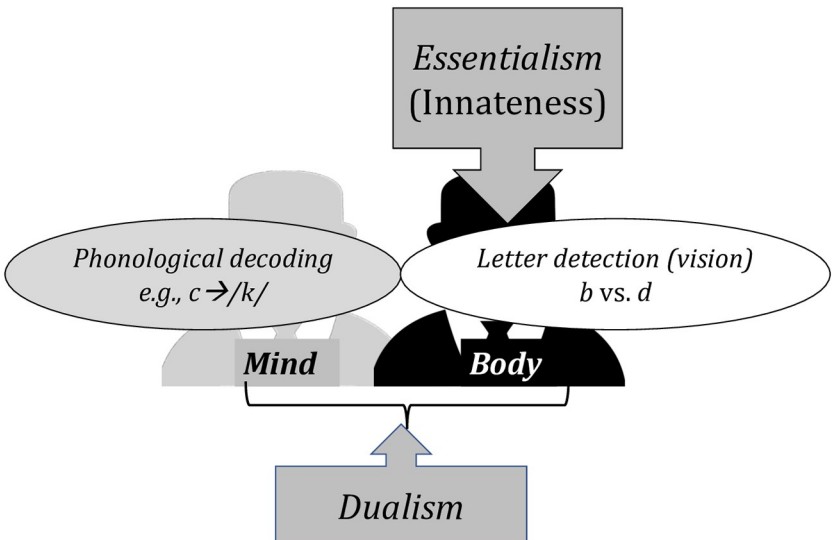

**Fig 2. The role of Dualism and Essentialism in reasoning about dyslexia.**

learned, people might also conclude that *phonological decoding is not associated with the body, including the brain*.

Altogether, the interaction between Dualism and Essentialism suggests several predictions about laypeople's attitudes towards dyslexia. First, we predict that people should be more likely to align dyslexia with visual difficulties (compared to troubles with phonological decoding). Second, people should tend to "essentialize" visual difficulties—they should consider visual difficulties as more likely to be biological, innate, immutable, and embodied (relative to phonological decoding). Third, since beliefs about the innate origins of dyslexia arise from the perceived anchoring of its symptoms in the body, we further predict that people should be more likely to essentialize *all* symptoms of dyslexia when their embodiment becomes salient. This can happen either because (i) people spontaneously anchor the symptom in the body (e.g., they link vision to the eyes); (ii) because they are offered *evidence* that the symptom is embodied (e.g., it "shows up" in the brain); or (iii) because the experimental manipulation renders the mind-body link salient (e.g., via priming). These predictions are summarized in (1).

1. *Predicted intuitive attitudes towards dyslexia*

   a. Dyslexia is more likely to arise from difficulties with vision (compared to troubles with phonological decoding).

   b. People should consider visual symptoms as more severe, immutable, and innate (relative to phonological decoding), in line with Essentialism.

   c. People should be prone to essentialize symptoms that they consider embodied; either because

      i. People spontaneously anchor the symptom with a particular bodily organ (e.g., with the eyes).

      ii. People are given evidence that the symptom is embodied (e.g., it "shows up" in the brain).

      iii. The experimental context renders the mind-body link salient.

The following experiments test these predictions. Experiment 1 first seeks to characterize laypeople's attitudes towards dyslexia and its origins more precisely. To this end, we examine whether attitudes towards the symptoms of dyslexia—as a visual or decoding disorder—are linked to their tendency to anchor the disorder in one's essence (i.e., to its innateness, biological status, and immutability). We also examine whether these essentialist attitudes are associated with the anchoring of the symptoms in the body (i.e., in the brain).

Having shown essentialist attitudes correlate with embodiment, in Experiments 2–3, we next explore whether the link is causal. Experiment 2 asks whether laypeople's attitudes towards dyslexia can be lawfully altered by manipulating people's attitudes about the anchoring of particular symptoms in the body (in the brain); in Experiment 3, we examine whether these attitudes can be altered by manipulating laypeople's attitudes about the mind-body links, generally.

## Experiment 1

Experiment 1 sought to unveil the link between laypeople's attitudes towards the symptoms of dyslexia and its perceived etiology. To this end, we asked participants to consider two individuals, John and Jack, who each suffer from reading difficulties. Both individuals read slowly,

and with great effort. Jack suffers from visual difficulties—he tends to confuse reversible letters like *b* and *p*. In contrast, John has troubles with decoding novel words—for him, *kat* is no more similar to an animal name than *vat*, for instance.

Participants were asked to evaluate the likelihood that each individual has a reading disorder, the severity of his condition, its prognosis (will the condition improve?), and its etiology—whether the condition originates from the person's effort and life experiences or from his biology, whether the condition affects the person's brain, whether it runs in his family, and whether it would likely transfer to a hypothetical genetic clone of the person, raised in a different social environment. Participants responded to each of these questions using a 1–7 scale.

We reason that, if people are more likely to consider visual symptoms as embodied, then visual symptoms should be considered as more likely to affect the brain (relative to phonological decoding).

If people further believe that dyslexia arises from one's embodied biological essence, then visual symptoms should be more likely to suggest a reading *disorder*, they should be more strongly associated with a *biological* causes, the disorder should be more likely to be considered *innate* (e.g., more likely to affect a family member and a genetic clone), and possibly, the disorder should be more severe and immutable.

Finally, if essentialist reasoning is triggered by embodiment, then responses to whether the disorder affects the brain should correlate with its perceived heritability, severity, and biological (rather than environmental) origin.

## Methods

**Participants.** Forty participants took part in this experiment. The selection of sample size was informed by related research [35]. Based on those previous results, we determined that the selected sample size is sufficient to yield a large effect size (.80) with a probability of .80 and an alpha level of .05.

Participants in this and all subsequent experiments were recruited from Amazon Mechanical Turk. They were all adult native English speakers who were reportedly free of language and reading disorders and had not taken any advanced courses in psychology (beyond an introductory course). Participants in all experiments were compensated $0.30 for their time.

Participants in Experiment 1 had also reportedly not taken any advanced courses in linguistics (100%), and many had not taken advanced courses in biology (83%). Of these participants, 23% reported their highest completed level of education to be high school, 55% as college, 23% as a graduate school program, and 0% as completing none of the above education.

To be included in the sample, participants had to further provide a coherent explanation for their response; this requirement was adopted in order to eliminate bot responses. To obtain the target sample (e.g., 40 participants), we ran a larger sample of 150% desired size (e.g., 60 participants), and selected the first 40 participants who provided a sufficient justification. Sufficient justifications were assessed liberally—we accepted any justification as long as it addressed the question and was not copied verbatim from the vignette (to avoid bot responses). The selection of the first N participant was done strictly by their order of their enrollment in the study, and it was blind to their responses. This screening procedure was applied to Experiment 1 & 3; in Experiment 2, the screening question was not displayed due to a technical error.

This study was approved by the Institutional Review Board at Northeastern University (#17-05-05). All participants signed an informed consent.

**Materials and procedures.** The materials consisted of two matched vignettes, each featuring one of two individuals—John and Jack—who suffer from reading difficulties.

John has difficulties in phonological decoding: John's reading is slow and deliberate, and he has specific difficulties with reading novel words. People were told that, when a typical reader is presented with the novel word *kat*, they immediately recognize that it sounds like the name of an animal. John, however, does not. For him, *kat* is no more similar to an animal name than vat, for instance. In contrast, Jack suffers from visual difficulties. When typical readers see the letters *b* and *d* they can readily tell them apart, even when presented for only a brief moment. Jack, however, fails to discriminate between these letters.

Participants read each vignette in a counterbalanced order. After each vignette, participants were asked to address eight questions (on a 1–7 scale; 1 = highly unlikely/strongly disagree/very minimal; 7 = highly likely/strongly agree/very severe): (1) Do you think it is likely that John's symptoms are indicative of a reading disorder? (2) How severe is the person's condition; (3) Does the person's difficulty result from their own actions, attitudes, and life experiences? (4) Does the difficulty originate from his biological makeup? (5) How likely is it that the person could improve his symptoms himself (e.g., by increasing his attention, trying harder, etc.)? (6) How likely is it that the difficulty affects the person's brain? (7) How likely is it that the difficulty runs in the person's family? (8) How likely it is that the difficulty would transfer to a genetic clone of the person, raised in an entirely different family and environment? (for the full vignettes, see, S1 Appendix in S1 File).

## Results

Fig 3 presents people's responses to "visual" and "decoding" symptoms. In this and all subsequent figures, error bars are 95% confidence intervals for the difference between the means.

To ensure that participants indeed considered these symptoms as indicative of dyslexia, we first compared the mean response against the scale's neutral midpoint (for results, see S1 Table in S1 File). Our main interest, however, is in responses to visual vs. decoding symptoms. These means were compared using matched-pairs t-tests.

Results showed that, compared to difficulties with decoding, "visual" symptoms were considered as more likely to indicate a reading disorder ($t(39) = 2.55$, $p = .01$, $d = .50$) and as more severe ($t(39) = 2.48$, $p = .02$, $d = 0.44$). However, the two symptoms—visual and decoding, did

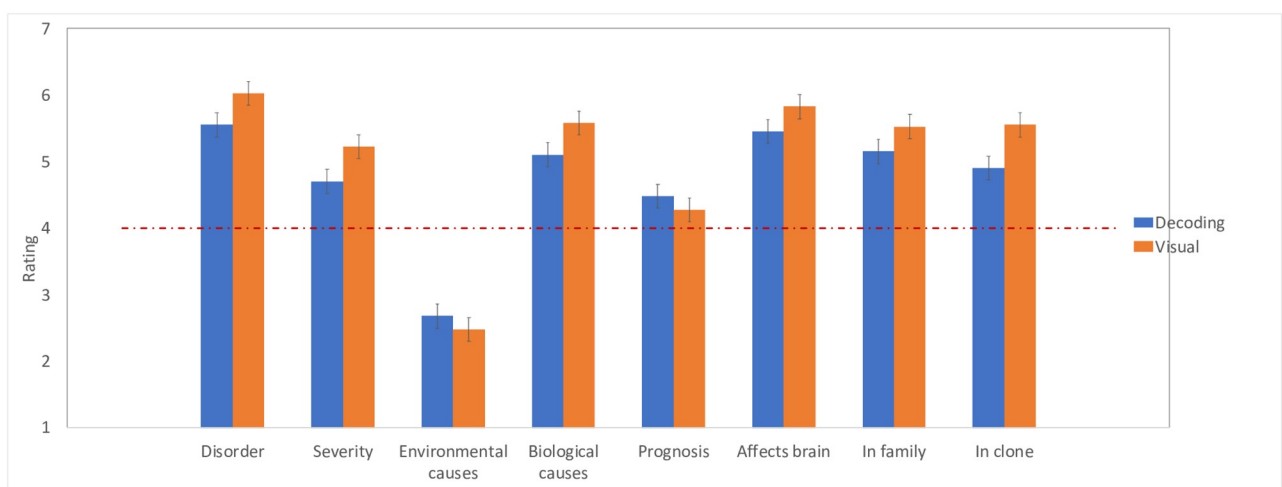

**Fig 3. The evaluation of reading disorders associated with visual and decoding difficulties.** Error bars in this and all subsequent figures are 95% confidence intervals for the difference between the means.

not differ with respect to environmental causes (t(39) = 1.02, p = .32, d = .13) or their prognosis (t(39) = 1.19, p = .24, d = .14).

Turning to the biology of the disorder, here, people believed that visual symptoms were more likely to result from biological causes (t(39) = 2.65, p = .01, d = 0.45), and more likely to affect the brain (t(39) = 2.20, p = .03, d = .37). Additionally, visual symptoms were considered more likely to run in the patient's family (t(39) = 2.15, p = .04, d = 0.40), and to transfer to his genetic clone (t(39) = 4.33, p<.0001, d = 0.59).

We next examined whether people's attitudes towards symptoms are linked to their perceived embodiment, gauged by their perceived instantiation in the brain (collapsing over symptom type). We found that brain symptoms were positively associated with innateness. The more likely the condition was to affect the brain, the stronger was its perceived tendency to "run in the family" (r(38) = .59, p<.0001) and to transfer to a genetic clone (r(38) = 0.54, p<.0005). The effect of the brain did not correlate with the likelihood of the symptoms to improve (r(38) = -.10, p = 53). However, people associated symptoms that affect the brain positively and significantly with a disorder (r(38) = .64,<.0001), with severity (r(38) = .52,<.0001), and with a biological cause (r(38) = 0.76,<.0001). In contrast, symptoms that affect the brain were negatively linked to experiential factor—this last correlation was marginally significant (r (38) = -.30, p = .06).

## Discussion

Experiment 1 sought to determine whether people's attitudes about the symptoms of dyslexia —as a visual or a decoding disorder—are linked to their attitudes about its severity and etiology.

Results showed that people considered visual difficulties as more likely to indicate a reading disorder, as more severe, more likely to affect the brain and to arise from a biological and hereditary causes (relative to phonological decoding). Although the prognosis of visual and decoding symptoms did not differ reliably, the prognosis of phonological decoding (but not visual symptoms) was higher than the scale's midpoint, suggesting that people thought that, unlike visual confusions, phonological symptoms can improve (for full results, see S1 File).

These results are predicted by the hypothesis that people are more likely to essentialize visual symptoms *because* they perceive them as more strongly anchored in the body (relative to phonological decoding). In line with this proposal, participants in Experiment 1 considered visual symptoms not only as innate and severe but they were also more strongly linked to the brain (relative to phonological decoding symptoms). Moreover, we found that, the stronger the association of the symptoms with the brain, the more likely people were to consider the symptoms as indicative of a disorder, innate, biological, and severe.

Experiments 2–3 move to examine whether the link between the embodiment of the symptoms and their innateness is ***causal***. If embodiment promotes essentialist thinking, then it should be possible to trigger essentialist thinking by manipulating the embodiment of dyslexia symptoms. Experiment 2 tests the essentialist predictions that "if it's in the body, it's innate". To this end, we assess whether people are indeed more likely to perceive dyslexia symptoms as inborn when they are given evidence that these particular symptoms manifest in the brain (compared to when the same symptoms are diagnosed behaviorally).

In Experiment 3, we examine whether the perceived innateness of symptoms is affected by attitudes about the mind-body links, generally. To this end, we primed people to consider bodies and minds as either distinct or linked (in line with Dualism vs. Physicalism, respectively). We asked whether the manipulation of intuitive Dualism affects the laypeople's attitudes towards dyslexia.

## Experiment 2

Experiment 2 examined whether laypeople's attitudes towards the etiology of a reading disorder can be manipulated by providing them with evidence that its symptoms manifest in the body, specifically, the brain.

To this end, participants were presented with the cases of two pairs of identical twins. Each pair exhibited precisely the same set of symptoms—either a difficulty with decoding or with visual letter reversals. In each case, the twins underwent a test designed to diagnose their condition, and the results of the test suggested that they exhibited a reading abnormality. One twin, however, was administered a behavioral test whereas his brother received a brain evaluation (as in our previous research on psychiatric disorders, [44]).

These two tests were exactly matched with respect to their diagnosticity—all they suggested was whether or not the person had a reading disorder. The brain test, however, presented participants with explicit evidence that the disorder manifests in the physical body, whereas the behavioral test did not offer such information explicitly. For the Dualist, this scenario thus leaves open the possibility that behavioral symptoms only affect the ethereal mind, whereas the brain symptoms are demonstrably embodied.

If embodiment triggers essentialist thinking, then people should be more likely to "essentialize" symptoms that they perceive as embodied—they should consider them as more likely to be innate (more likely to be affect family members and clones), biological (rather than environmentally cased), and immutable (having worse prognosis and being more severe), in line with the results of Experiment 1. But since in Experiment 2, the task calls attention to the test (as the twins are explicitly contrasted on this dimension) and deemphasizes the symptom (as the twins are matched on that dimension), in Experiment 2, we expect the effect of embodiment to be driven primarily by the test (brain vs. behavior) rather than by the symptom (visual vs. phonological). Accordingly, we expect people to consider disorders that are diagnosed by the brain test as more likely to be biological, innate, and immutable. Additionally, the perception of the disorder as affecting the brain should correlate with its perception as biological, innate, and immutable.

### Methods

**Participants.**　Forty participants took part in the experiment. Participants in Experiment 2 had reportedly not taken any advanced courses in linguistics (100%), and many had not taken advanced courses in biology (78%). Of these participants, 13% reported their highest completed level of education to be high school, 53% as college, 33% as a graduate school program, and 3% as completing none of the above education.

**Materials and procedure.**　Each participant was presented with two vignettes (see, S2 Appendix in S1 File). Each such vignette featured a pair of twins who each suffer from reading difficulties—either a difficulty in phonological decoding, or visual letter-reversals. For each pair, both twins suffer from the same difficulty (phonological or visual); each such difficulty was described as in Experiment 1.

The twins both see a reading specialist friend, who urges them to get screened for reading disorder. The friend further explains how their particular difficulties can lead to a reading disorder. For the phonological decoding vignette, the twins are told that typical readers instinctively rely on abstract rules that match letters with sounds; when these rules are impaired, this can lead to a reading disorder. Similarly, for the visual vignette, the twins are told that when people learn to read, they develop the ability to discriminate between mirror-symmetrical letters, such as *b* and *d*. The friend worries that the twins might have a disorder that impairs this visual process.

To diagnose these disorders (phonological and visual), the twins undergo an appropriate assessment of either their phonological decoding of novel words or of brief displays of similar letters, such as *b* and *d*. One twin member, however, is diagnosed by a brain test whereas his brother is administered a behavioral test.

Both tests were strictly normed, and their results, in both cases, suggested abnormality. These test outcomes, then, effectively addressed the first of the eight questions from Experiment 1, so in Experiment 2, we did not ask participants to evaluate whether the symptoms indicated a disorder. Participants, however, were asked to evaluate (on a 1–7 scale) the remaining seven questions, namely, the severity of the disorder, its causes (biological and environmental), its prognosis, its tendency to run in the person's family, and its potential to transfer to its clone. The two symptoms (visual and decoding) and two tests (brain vs. behavior) were counterbalanced for order in four lists, and each list was assigned to 10 participants. Participants answered all seven questions about a single individual separately, before moving on to the next individual.

## Results

Fig 4 presents participants' ratings for visual and decoding symptoms. An inspection of the means suggests that in nearly all cases, responses differed systematically depending on the method of diagnosis—a brain- or a behavioral test.

We next assessed these results by means of 2 Symptoms (Decoding/Visual) x 2 Test (Brain/Behavior) fully repeated-measures ANOVAs, applied to each of the seven questions. In what follows, we list all significant results; for ease of exposition, we also provide the results in S2 Table in S1 File.

Results showed that the effect of ***test*** reliably modulated the perceived causes of the disorder —environmental ($F(1,39) = 5.68$, $p = .02$, $\eta^2 = 0.13$) and biological ($F(1,39) = 10.68$, $p<.003$, $\eta^2 = 0.22$), its prognosis ($F(1,39) = 8.49$, $p = .01$, $\eta^2 = 0.18$), and its tendency to transfer to a genetic clone ($F(1,39) = 8.30$, $p<.007$, $\eta^2 = 0.18$). The effect of test type on the propensity to affect the brain was marginally significant ($F(1,39) = 3.58$, $p = .07$, $\eta^2 = 0.08$). In all cases, the effect of test was not further modulated by the symptom type. Symptom type only affected perceived severity, as decoding difficulties were considered more severe than visual difficulties, a result that was marginally significant ($F(1,39) = 4.22$, $p = .05$, $\eta^2 = 0.10$).

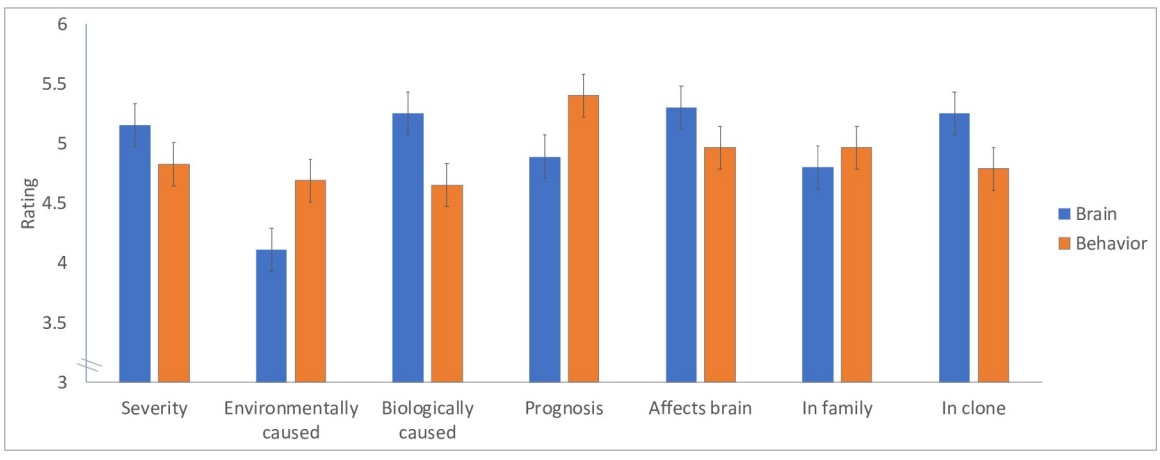

**Fig 4. The evaluation of reading disorders diagnosed by a behavioral and brain tests.**

**Table 1. Correlation between the perceived tendency of dyslexia to affect the brain and its perceived origins and prognosis.**

| | Severity | Experience | Biology | Will improve? | In family? | In clone? |
|---|---|---|---|---|---|---|
| **Brain test** | **0.47**$^\$$ | 0.18 | **0.65**$^\#$ | -0.07 | **0.77**$^\#$ | **0.66**$^\#$ |
| **Behavioral test** | 0.26 | **0.35**$^*$ | **0.72**$^\#$ | 0.08 | **0.79**$^\#$ | **0.66**$^\#$ |

Note:

$^*$ p<.05;

$^\$$ p<.01;

$^\#$ p<.001.

These results indicate that symptoms that manifested in the brain test were perceived as more likely to be biologically- and innately (as opposed to environmentally) based. Symptoms detected by the brain test were also associated with poorer prognosis, and as would be expected, people also tended to consider the disorder as more likely to affect the brain.

**Correlational analysis.** To further gauge the link between embodiment and essentialist thinking, we next correlated responses to the "affects brain" question with all other questions, collapsed over symptoms.

Results (see Table 1) showed that the perceived propensity of the symptom to affect the brain correlated positively with its perceived tendency to run in patients' families, to transfer to a genetic clone, and to arise from a person's biological makeup; in all cases, this was the case regardless of test.

Other effects of embodiment, however, were test-dependent. When the disorder was diagnosed by the brain test only, people considered disorders that affect the brain as more severe. In contrast, when the test was behavioral, people considered disorders that affected the brain as more likely to arise from one's experience.

## Discussion

Experiment 2 shows that when people were given evidence that dyslexia symptoms are embodied—they "show up in the brain"—people considered the symptoms as more likely to be innate, biologically- (as opposed to environmentally) caused, and as having poorer prognosis, compared to when the same symptoms were diagnosed behaviorally. As expected, people also tended to view symptoms that "show up" in the brain as more likely to affect the brain.

These results demonstrate for the first time that people's attitudes concerning the origins of dyslexia are *causally* linked to their manifestation in the brain. These results agree with our previous findings for typical psychological traits and psychiatric disorders [42, 44]—in all cases, people consider brain symptoms as more likely to be innate. To be clear, the presumption that embodiment implies innateness is false. Modern science suggests that all psychological symptoms—both innate and acquired—manifest in the brain. But laypeople incorrectly presume that symptoms that manifest in the brain are immutable and inborn.

In addition to these causal links, the correlational analysis further showed that the perceived anchoring of dyslexia symptoms in the brain was associated with their attribution to an innate, biological source. Moreover, for symptoms diagnosed by the brain test, the perceived anchoring in the brain was associated with greater severity, whereas for those diagnosed behaviorally, the effect of the symptom on the brain was associated with one's experience.

Symptom type, in this experiment, generally did not affect the responses; the only exception was an unexpected tendency of an increase in the severity of decoding relative to visual difficulties. Upon closer inspection, it appears that decoding was perceived as more severe only when it manifested in the brain test (marginally so, t(39) = 1.87, p = .07; d = -0.29; for the

behavioral test: t(39) = 1.10, p = .28; d = -0.11). We speculate that, since (as shown in Experiment 1), decoding is not typically associated with biology, when decoding difficulties demonstrably do affect the brain, people conclude that the underlying disorder must be particularly severe.

Overall, the results of Experiment 2 show that, when participants are provided evidence that dyslexia symptoms are embodied, people tend to essentialize them—they consider the symptoms as more biological, immutable, and innate. These results converge with the findings from Experiment 1, where participants exhibited similar essentialist attitudes towards visual symptoms—symptoms that they naturally consider as embodied. In Experiment 2, however, essentialist attitudes were unaffected by symptom type, probably because (a) participants were explicitly informed that, regardless of symptom, *all* patients suffered from a known reading disorder; and (b) participants were specifically asked to compare the outcome of the two tests to each other, rather than to contrast the visual and decoding symptoms (as they did in Experiment 1). Experiment 3 further addresses the contrast between visual and decoding symptoms, and the effect of the mind-body links on their perceived severity.

## Experiment 3

Experiment 3 had two goals. The first was to demonstrate that, when participants are prompted to compare visual and decoding symptoms, they once again consider visual symptoms as more severe, affecting the brain, and innate, replicating the findings of Experiment 1.

Second, we sought to further demonstrate that attitudes towards the disorder are caused by the perceived links between the mind and the body. Our approach, here, was complementary to the one we adopted in Experiment 2. Experiment 2 informed participants about the physical basis of *specific symptoms* by manipulating their propensity to manifest in a brain test. Experiment 3, by contrast, manipulated participants' attitudes towards the mind-body *generally*.

The procedure included three parts (as in [35, 58]). First, we had participants read a short essay on the link between the body and mind. One group of participants read an essay describing the mind as distinct from the body, in line with mind-body Dualism; a second essay administered to another group discussed the mind and body as one and the same, in line with Physicalism. To gauge the effectiveness of the priming manipulation, we next asked participants rate their perceptions of the mind-body distance. Finally, we had participants read two vignettes—one describing the symptoms of a patient who suffers from a decoding difficulty; another described visual symptoms related to letter reversals. Participants were asked to evaluate the likelihood of a disorder, its severity, and etiology, as in Experiment 1.

We reasoned that, if the distinction between visual- and phonological symptoms of dyslexia arises from the perception of the phonological symptoms as relatively ethereal, distinct from the body, then this distinction is more likely to emerge when participants are prompted to think about the body and mind as distinct than as one and the same (i.e., the Dualist relative to the Physicalist condition).

Additionally, now that participants are invited to contrast the visual and decoding symptoms, and the status of the diagnosis is once again ambiguous, we expected participants to view visual symptoms as more severe and biologically based, in line with Experiment 1.

### Methods

**Participants.**   This experiment included two groups of participants (N = 60 per group). One group was assigned to the Dualism condition, another was assigned to the physicalism condition. This sample size was determined according to the one used in [58].

Participants in Experiment 3 had reportedly not taken any advanced courses in linguistics (100%), and many had not taken advanced courses in biology (90%). Of these participants, 33% reported their highest completed level of education to be high school, 55% as college, 12% as a graduate school program, and 1% as completing none of the above education. Experiment 3 required the same requirement as Experiment 1 for participants to include a coherent explanation to a question in order to be included in the sample. To obtain the target sample (e.g., 60 participants per Dualism/Physicalism condition), we ran a larger sample of 150% desired size, and selected the first 60 participants from each condition who provided a sufficient justification. The selection of the first N participant was done strictly by their order of their enrollment in the study, and it was blind to their responses.

**Materials and procedures.** The experiment had three parts (as in [35, 58]). First, participants read a passage discussing the link between body and mind; in the physicalist condition, participants were told the mind and body were one and the same; the dualist condition described them as distinct.

Second, participants were presented with seven diagrams, each depicting the distance between body and mind as the distance between two circles; participants were asked to choose which diagram best presented the relationship between the body and the mind (on a 1–7 scale; higher ratings indicated greater separation between mind and body).

Finally, participants were presented with the case of two patients who suffer from either decoding or visual difficulties (as described in Experiment 1). The order of the decoding and visual difficulties was counterbalanced across participants. Participants were asked to rate (on a 1–7 scale) whether the person had a reading disorder, its severity, and etiology, using the eight questions from Experiment 1. For the full materials, see, S3 Appendix in S1 File.

## Results

**a. Manipulation check.** Before inspecting the effect of the priming manipulation on attitudes towards dyslexia, we first sought to establish that our manipulation was effective. To this end, we compared participants' perception of the mind-body distance in the physicalist and dualist condition using a two-sample t-test. Results suggested that participants perceived the mind-body distance as significantly larger in the Dualist condition (M = 3.63, SD = 2.15) compared to the physicalist condition (M = 1.27, SD = 0.78; t(119) = 7.99, p<.0001). Thus, the priming manipulation did alter participants' perception of the mind-body distance.

**b. Attitudes towards dyslexia by the mind/body manipulation.** To evaluate the effect of the perceived mind-body divide (as determined by the priming manipulations) on attitudes towards dyslexia, we next submitted the responses to each of the eight questions to a separate 2 Priming (dualism/physicalism) x 2 Symptom (visual/decoding) ANOVA. In this analysis, Priming is manipulated between subjects; Symptom is manipulated within subjects. Below, we summarize all significant results; we also provide all ANOVA results in, S3 Table in S1 File.

We found that priming significantly modulated only people's attitudes of the severity of the disorder; the Priming x Symptoms interaction was marginally significant (F(1,118) = 3.94, p = .05, $\eta^2$ = 0.03) (see Fig 5).

Planned contrasts showed that in the Dualism condition, people rated visual symptoms as more severe than decoding difficulties (t(215.46) = 2.55, p = .01), in line with the results of Experiment 1. But when participants were led to consider the mind and body as one and the same, in the physicalist condition, responses to the symptoms did not differ reliably (t(215.46)<1).

For the remaining seven questions, there was no reliable effect of Priming (all p>.26) nor was the Priming x Symptom interaction significant (all p>.34). However, Symptom type reliably affected responses (see Fig 6).

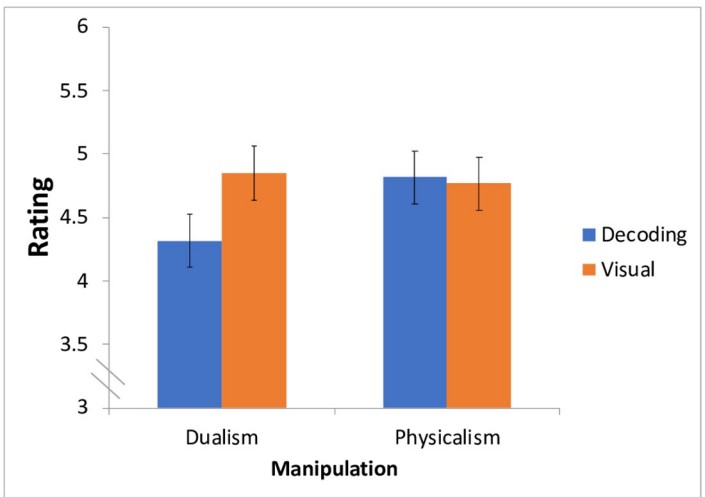

**Fig 5. The effect of the mind/body manipulation on the perceived severity of dyslexia symptoms (phonological decoding and visual).**

Specifically, people considered visual symptoms as more indicative of a disorder ($F(1,118)$ = 36.74, p<.0001 $\eta^2$ = 0.24), as more indicative of biological ($F(1,118)$ = 7.54, p<.008, $\eta^2$ = 0.06), but not environmental ($F(1,118)$ = 17.67, p<.0001, $\eta^2$ = 0.13) origin, as having worse prognosis ($F(1,118)$ = 6.48, p = .01, $\eta^2$ = 0.05), as more likely to affect the brain ($F(1,118)$ = 4.71, p = .03, $\eta^2$ = 0.04), to run in a person's family ($F(1,118)$ = 10.10, p<.002, $\eta^2$ = 0.08), and to transfer to a clone ($F(1,118)$ = 7.01, p = .01, $\eta^2$ = 0.06). Thus, replicating Experiment 1, participants in Experiment 3 considered visual symptoms as more severe, as more likely to affect the brain, and to be heritable.

**Correlational analysis.** To further evaluate the association between the perceived embodiment of the symptoms and attitudes towards dyslexia, we correlated responses to the "affects the brain" question with responses to all other questions (collapsed over symptom type). We found that "affects the brain" question was positively associated with the perception of the symptoms as innate (as likely to run in the family, and transfer to one's genetic clone) and as

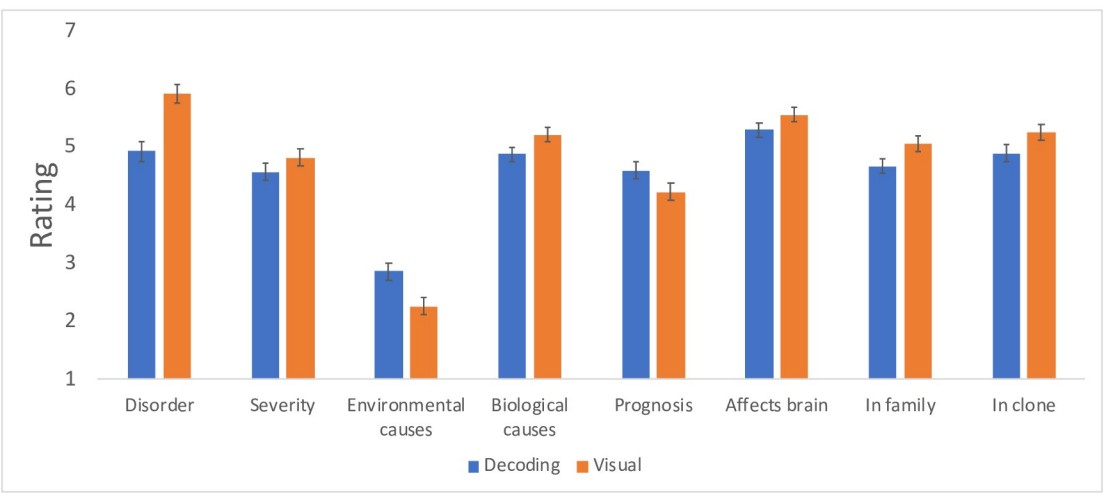

**Fig 6. The perception of reading disorders associated with visual and decoding difficulties.**

**Table 2. Correlation between the perceived tendency of dyslexia to affect the brain and its perceived origins and prognosis.**

|  | *Disorder* | *Severity* | *Experience* | *Biology* | *Will improve?* | In family | In clone? |
|---|---|---|---|---|---|---|---|
| **Dualism** | 0.44[#] | 0.51[#] | -0.45[#] | 0.70[#] | -0.42[#] | 0.43[#] | 0.62[#] |
| **Physicalism** | 0.19 | 0.30* | -0.46[#] | 0.60[#] | -0.40[$] | 0.53[#] | 0.44[#] |

Note:

* p<.05;

$ p<.01;

# p<.001.

more severe, and this was the case regardless of priming (see Table 2). Likewise, people in both conditions positively associated symptoms that affect the brain with biology, and negatively with an experiential source; once again, this was regardless of the priming manipulations.

The Dualism and Physicalist conditions, however, differed in so far as symptoms that were perceived to affect the brain were significantly associated with a greater likelihood of disorder only in the Dualism, but not in the Physicalist conditions.

## Discussion

Experiment 3 has yielded two major results. First, participants considered visual symptoms as more indicative of a reading disorder, and they were more likely to attribute them to an inborn biological condition that affects the brain relative to phonological difficulties (in line with Experiment 1). Additionally, participants in Experiment 2 associated visual symptoms with poorer prognosis.

Second, attitudes towards dyslexia were modulated by the mind-body link. Participants who were primed to think about the mind and body as distinct viewed decoding symptoms as less severe than visual symptoms, in line with their baseline responses in Experiment 1 (in the absence of priming). But when participants were prompted to consider body and mind one of the same, visual and phonological symptoms were considered as equally severe. These results suggest that the greater perceived severity of visual symptoms arises from the perception of the mind as distinct from the body. Moreover, the fact that responses in the Dualist condition did not differ from the baseline condition (in Experiment 1) suggests that people are inherently Dualists (in line with [35]).

As in previous experiments, these participants tended to essentialize symptoms that they associate with the brain. Consequently, the perception of symptoms as linked to the brain was positively associated with the likelihood that the disorder is innate (i.e., that it runs in the family and in one's clone), severe and biological, and negatively associated with one's experience and with a better prognosis.

One of this correlation, however, was specifically affected by the mind-body links. Participants in the Dualist condition associated brain symptoms with higher likelihood of a disorder (as in Experiment 1); this, however, was not the case in the Physicalist condition. These results suggest that participants' tendency to "essentialize" brain symptoms arises, in part, from their perception of such symptoms as distinct from the body.

These results agree with our previous findings, demonstrating that the perception of psychological traits as innate can be altered by manipulating participants' perception of the mind/body link [35]. The present results, however, are the first to show that the salience of the mind-body link can affect attitudes towards dyslexia.

## General discussion

This research sought to explore public attitudes towards dyslexia and shed light on their origins. Reading science suggests that dyslexia is a hereditary brain disorder that typically compromises phonological decoding [1, 2]. Laypeople, however, typically attribute dyslexia to visual confusions [3–5, 17], and they also have troubles grasping that dyslexia is a biogenetic condition (e.g., [4, 5]).

Here we asked whether these public misconceptions could arise, in part from intuitive psychology—specifically, from the conflict between intuitive Dualism and Essentialism. We hypothesized that people fail to link dyslexia to phonological decoding because they correctly assume that dyslexia is a hereditary disorder, and per intuitive Essentialism, they assume that one's innate immutable essence must reside in one's body (for review, [39]). Per intuitive Dualism, however, people incorrectly consider phonological decoding as ethereal, distinct from the body [24], so people cannot fathom how phonological decoding could be affected by an innate biological condition. Since visual confusions are readily perceived as embodied, laypeople conclude that dyslexia is far more likely to arise from visual confusions, which they link to one's innate essence.

Altogether, then, we suggest that people only essentialize dyslexia symptoms that they can anchor in the body—either because (a) the symptom is linked to a bodily organ (e.g., eyes, for vision), because (b) people are provided evidence for embodiment, or because (c) the experimental manipulation underscores the mind-body link. Experiments 1–3 explore these three predictions, respectively.

Experiment 1 showed that attitudes towards the etiology of the disorder and its prognosis indeed depend on its symptoms—visual confusions (e.g., of *b/d*) vs. difficulties with phonological decoding. We hypothesized that visual symptoms should be readily linked to the body, and consequently, people should be more likely to essentialize them. In line with this prediction, people considered visual symptoms as more indicative of reading disorder, more severe, and more likely to affect the brain, and more likely to arise from a biological and hereditary cause relative to phonological decoding. Additionally, the tendency to essentialize symptoms (i.e., to consider them as indicative of a disorder, biological, and severe) correlated with their perception as embodied (i.e., as anchored in the brain). The tendency to consider visual symptoms as biological, innate, and severe is in line with their view as indicative of one's innate, immutable biological essence. As such, these results open up the possibility that people essentialize visual symptoms *because* these symptoms are embodied.

Experiment 2–3 moved to investigate whether the link between the embodiment of the symptoms and essentialist reasoning is causal. If it is, then it should be possible to alter laypeople's essentialist attitudes towards the symptoms by manipulating their link to the body.

In Experiment 2, embodiment was manipulated by providing evidence that the symptoms affect the body (i.e., that they "show up" in the brain); the (control) behavioral test indicated a disorder without providing explicit evidence that the symptoms were embodied. Results showed that, when the disorder manifested in the brain, participants considered the disorder as more biologically based and inborn compared to when the same symptoms were diagnosed behaviorally, and in addition, the prognosis of the disorder seemed poorer. Thus, people tended to interpret symptoms affecting the brain as innate and immutable. In addition, the stronger the anchoring of the symptom in the body (brain), the stronger was their perception as innate and biological, and when the diagnosis was offered by the brain test, the association of the symptom with the brain was linked with their view as more severe.

Experiment 3 showed that people's attitudes towards the disorder were further modulated by the salience of the mind body link. Here, participants who were primed to view the body

and mind as distinct considered visual symptoms as more severe than decoding difficulties, as did participants in Experiment 1 (who received no priming manipulation). But when participants were prompted to consider the body and mind as one and the same, the perceived severity of visual symptoms was eliminated. These results suggest that the greater severity of visual symptoms (potentially, a telltale sign of Essentialism) arises from the view of the mind and body as distinct.

The salience of the mind-body link also affected attitudes about the likelihood of the disorder, in Experiment 3. While participants in the Dualist conditions associated symptoms that affect the brain with a reading disorder (in line with the findings from Experiment 1), this was no longer the case when participants considered the body and mind as one and the same (in the physicalist condition). In both conditions, however, symptoms that were anchored in the body (brain) were associated with the likelihood that the disorder is innate (i.e., that it runs in the family and in one's clone), severe and biological, and negatively associated with one's experience and with a better prognosis.

Finally, Experiment 3 found that people considered visual symptoms as more indicative of a reading disorder, immutable (i.e., having worse prognosis), biologically based, and innate compared to decoding symptoms (replicating Experiment 1).

Altogether, then, Experiments 1–3 suggest that people are more likely to consider symptoms that are readily linked to the body as indicative of an innate, severe, and immutable reading disorder.

As noted earlier, laypeople's tendency to view dyslexia as a visual disorder can arise from multiple sources, including the fact that some individuals with dyslexia do indeed suffer from visual difficulties (e.g., [12, 13]). Moreover, since phonological decoding clearly requires instruction and practice, whereas visual processing arises spontaneously, it is only natural for the public to consider visual symptoms as more hereditary and immutable.

While these considerations can well account for some public attitudes, they are insufficient to capture the results. In particular, these accounts do not explain why people believe that visual symptoms are more likely to affect the brain (relative to phonological decoding), and why brain symptoms are considered more severe and hereditary. Furthermore, as we have shown, essentialist reasoning about dyslexia is causally linked to the embodiment of the symptoms generally; the tendency to essentialize visual symptoms is just a special case of this broader bias. The proposal that people are simply misinformed about the origins of dyslexia, or confused by the learning of phonological decoding fails to explain this broader role of embodiment. The conflict we have outlined between Dualism and Essentialism explains these facts.

Briefly, Essentialism is the intuitive belief that living things possess an inborn immutable essence (e.g., [25, 36]). While the notion of essentialism has been extended to capture reasoning about nonbiological kinds (e.g., [59–61]), there is evidence that, when people consider biological agents, not only do they presume that these kinds possess an underlying essence, but they further presume that the essence must be *embodied* (for review [39]). For example, young children believe that a dog inherits its color from a small piece of matter it received from its mother [62], whereas the essence of living things is localized in their center [40]. This proposal is further supported by recent results, suggesting that, when adults are provided explicit evidence that a given psychological trait is anchored in the body—either in the brain, in the face, or in the internal body, people are more likely to consider the trait in question as innate [35, 42].

Laypeople, however, are also Dualists—they consider the mind as immaterial, distinct from the body [24]) Past research suggests that, not only do people view the psyche as less material than the physical body [30–34], but they further view certain psychological traits as more

material than others [21, 35]. Sensory traits like vision are readily linked to the material body, as people can identify them with specific bodily organs [35]. But when it comes to abstract epistemic capacities, such as phonological decoding, here, the link to the physical body is not patent. Indeed, adults struggle to link typical epistemic capacities to the brain (relative to sensorimotor ones; [35]).

Since Essentialism demands that innate biological traits be in the body, whereas per Dualism, some traits are ethereal and disembodied, it follows that only certain psychological traits —those that are anchored in the body—can be essentialized. Like other visual traits, vision is naturally seen as embodied (in the eyes), so people are naturally prone to essentialize visual symptoms of dyslexia—they consider them as more likely to be biological, innate, and severe. But since essentialist reasoning is triggered by the perception of the symptoms as embodied broadly, people are likewise prone to essentialize all symptoms of dyslexia when their embodiment becomes salient—either by providing evidence that the symptoms "show up "in the brain (in Experiment 2) or by strengthening the mind-body links generally (in Experiment 3). Altogether, then, our results suggest that the perception of symptoms as embodied—either in the eyes (in Experiments 1& 3), or tin the body (in Experiments 2 & 3) promotes essentialist thinking.

The principles we have invoked to explain laypeople's attitudes towards dyslexia can further explain other attitudes towards the psyche—in both health and disease. In past research, we have shown that, when people reason about the origins of typical psychological traits, they tend to consider traits that they link to the body as innate [35, 42] and as indicative of one's essence [45].

Our present findings also converge with the conclusions emerging from the investigation of laypeople's attitudes towards psychiatric disorders (e.g., [54, 56, 63, 64]). As in the case of dyslexia, people perceive psychiatric symptoms as more immutable and severe when the symptoms are linked to the brain [65]. Moreover, people are more likely to view conditions such as depression and anxiety disorder as inborn when the symptoms are diagnosed by a brain test compared to when the same symptoms are diagnosed behaviorally [44]

Such observations offer additional evidence as to how neuroscience can seductively interfere with laypeople's reasoning [23]. A large literature shows that people consider brain explanation as more credible than behavioral explanations [23, 66–74]. Our present results contribute to this literature by showing that people further assign brain results special status in reasoning about the cognitive disabilities, and that these beliefs can be traced to intuitive psychological principles that govern reasoning about the links between body and mind and about the essence of living things.

Finally, the present results could help inform efforts to promote public understanding of dyslexia. Our finding show that people incorrectly interpret brain imaging as suggesting that the underlying disorder is an immutable destiny. To counteract such biases, it might be appropriate to explicitly inform the public of the proper interpretation of neuroimaging studies, and to underscore that dyslexia is highly malleable and treatable. We hope that by shedding light on such biases, our research could contribute to these efforts.

## Supporting information

**S1 File.**
(DOCX)

**S1 Data.**
(XLSX)

## Author Contributions

**Conceptualization:** Iris Berent.

**Data curation:** Melanie Platt.

**Formal analysis:** Melanie Platt.

**Project administration:** Melanie Platt.

**Writing – original draft:** Iris Berent.

**Writing – review & editing:** Iris Berent, Melanie Platt.

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
