## [Decision Letter · Decision Letter 0]

7 May 2021

PONE-D-20-39703

Public misconceptions about dyslexia: the role of intuitive psychology

PLOS ONE

Dear Dr. Berent,

Thank you for submitting your manuscript to PLOS ONE. After careful consideration, we feel that it has merit but does not fully meet PLOS ONE’s publication criteria as it currently stands. Therefore, we invite you to submit a revised version of the manuscript that addresses the points raised during the review process.

We look forward to receiving your revised manuscript.

Kind regards,

Norbert Maionchi-Pino, Ph.D

Academic Editor

PLOS ONE

Journal Requirements:

Please provide additional details regarding participant consent. In the ethics statement in the Methods and online submission information, please ensure that you have specified (1) whether consent was informed and (2) what type you obtained (for instance, written or verbal, and if verbal, how it was documented and witnessed). If your study included minors, state whether you obtained consent from parents or guardians. If the need for consent was waived by the ethics committee, please include this information.

Please include captions for your Supporting Information files at the end of your manuscript, and update any in-text citations to match accordingly. Please see our Supporting Information guidelines for more information: http://journals.plos.org/plosone/s/supporting-information.

Additional Editor Comments :

Dear Pr. Berent

Thank you for submitting your manuscript to PO. Please first let me be sorry for the time to make a decision. Due to the pandemic situation, we experienced difficulties in finding available reviewers with expertise in the field of your manuscript. Fortunately, I have now received two detailed and insightful reviews for your manuscript. I have also read it myself.

After careful consideration, I feel that it has merit but does not fully meet PO’s publication criteria as it currently stands. Both reviewers suggest major revisions and I agree with their comments. I sincerely think this manuscript could be suitable for a publication in PO since it addresses a quite unexplored aspect of dyslexia. I therefore would like to invite you to submit a revised version of the manuscript that addresses the points raised during the review process.

From my point of view, I would insist on the fact that you need to improve your Introduction section; my major concern lies in a “disconnection” between paragraphs which makes reading difficult to follow. I am not used with some “philosophical” terms that you mentioned in your manuscript, so I think it contributes to make my own reading laborious. Maybe I am misleading but I have the feeling that you skipped some aspects related to dyslexia in your theoretical framework (also see comments made by Reviewer #1).

I also mostly agree with reviewer's #2 comments that your Results section needs an in-depth justification; indeed, some results were just at .05 with weak n²p. Some of your conclusions could be too straight and confident. Maybe could you qualify some conclusions?

To conclude, I have no important comments to add to this decision since both reviewers have provided a number of thoughtful suggestions related to the Introduction, Method and Results sections, and as both are more expert in this field than I am, I will defer to them on this score.

I would therefore offer the opportunity to submit a revision of the manuscript. I am sure my decision is in line with the major points raised by the reviewers.

Thank you for considering PO for you manuscript. We look forward to receiving your revised manuscript.

Regards,

N. Maïonchi-Pino

Reviewers' comments:

Reviewer's Responses to Questions

**Comments to the Author**

1. Is the manuscript technically sound, and do the data support the conclusions?

Reviewer #1: Partly

Reviewer #2: Partly

2. Has the statistical analysis been performed appropriately and rigorously? 

Reviewer #1: Yes

Reviewer #2: No

3. Have the authors made all data underlying the findings in their manuscript fully available?

Reviewer #1: Yes

Reviewer #2: Yes

4. Is the manuscript presented in an intelligible fashion and written in standard English?

Reviewer #1: Yes

Reviewer #2: Yes

5. Review Comments to the Author

Reviewer #1: This paper presents three studies that explore how the symptoms associated with dyslexia influence people’s perceptions of the condition. In Experiment 1, the authors examined how telling participants that a person is experiencing visual or decoding difficulties influences their beliefs about the severity and biological basis of the illness. In Experiment 2, they examined how the symptoms interact with whether the illness was diagnosed through a behavioral or brain test. In Experiment 3, they explore how the symptoms interact with people’s dualistic beliefs. The idea of dualistic beliefs influencing people’s reasoning about dyslexia is interesting, however many aspects of the paper need considerable revisions.

Major issues

1. The authors should consider expanding and reorganizing the introduction. Currently the introduction reads somewhat disjointed, with each paragraph presenting new information, but little connecting paragraphs or articulating the argument more explicitly for the reader. Additionally, the authors used jargon terms in the introduction without defining or explaining them. Some examples include “deep orthography,” “intuitive psychology,” “intuitive biases,” “dualism,” and “essentialism.” Explaining these terms would help a naïve reader understand the introduction. I also found it interesting that the introduction does not review literature on laypeople’s views on mental illness. The authors cite some of this work in the discussion, but some mention of how people reason about mental illness might be useful in the introduction. The authors should consider moving some of the information presented in Part 2 and the general discussion to the introduction as these sections articulate their argument more and do a good job at summarizing prior research.

2. One aspect that was not clear to me as a reader was the intended audience for this paper. If the audience is cognitive scientist interested in people’s concepts, then the authors should consider justifying why examining people’s views of dyslexia (specifically) might be useful. If the audience is dyslexia researchers, then more information on the discussion about how these biases might influence issues such as intervention choices or stigma might be needed. If they want the article to appeal to both groups, the authors should consider addressing both issues.

3. The authors should consider explaining how their review of the literature leads to their hypotheses. When I first read the hypothesis for Experiment 1 “biological disorders are more likely to affect sensory capacities than cognitive learning mechanisms” (pg. 4) I was not sure how that was related to the other issues discussed in the introduction. My guess is that this hypothesis is related to dualism, but this concept has not been introduced until a later paragraph.

4. I found the sample sizes in all the studies to be fairly small, particularly for researchers using online recruitment methods such as Mechanical Turk. In addition, the authors do not present a power analysis or a justification for their sample size. Even though Experiment 1 has only 40 participants, it has the benefit that Experiment 3 uses similar methods with a higher sample size (120 participants), effectively serving as a replication. However, there is no replication of Experiment 2, which also has only 40 participants, but was additionally testing for interactions. Given that there is no justification for the sample size of any of the studies, I have doubts on whether these studies were adequately powered to detect interactions. For example, some of the effects that were not significant in Experiment 1 were significant in Experiment 3 (which had a higher sample size). It would be useful to have a high-powered replication of the effects shown on this paper.

5. There should be more information about the actual materials and procedures of the study in the body of the manuscript. Currently, the actual stimuli are in the supplemental materials, but the authors don’t present a description of their methods in the Method section. Additionally, please make it clear in the description of study 2 that both of the twins in one vignette had either visual or decoding symptoms.

6. More information is needed as to what the author’s hypotheses were and how these hypotheses relate to their outcome measures. For example, one hypothesis presented is that visual disorders would be perceived as more biological. This hypothesis explains why the authors asked about environmental and biological causes, whether the disorder affects the brain, runs in the family, and a clone would have it. However, it is less clear why some of the other questions (e.g., severity, prognosis, indicative of a reading disorder) are being asked. A revised introduction could explain why these variables are being included, how they relate to their overall argument, and what the authors hypothesized (conversely, if these variables are exploratory, they should be labelled as such).

7. In page 7, the authors claim, “These results make it clear that not only are laypeople more likely to associate dyslexia with visual difficulties (rather than decoding difficulties), but that their attitudes towards the symptoms of the disorder and its etiology are linked.” The second part of the statement is support by their data as people who read the disorder was associated with visual impairments thought it was more likely for it to have biological causes. However, it is not clear to me which aspect of their data speaks to the first part of their statement. The authors never asked participants whether dyslexia was associated with visual or decoding difficulties. They did have a question asking whether the symptoms are indicative of a reading disorder (generally), not dyslexia (specifically). Given some of the work the authors review in the introduction, I can see why they make this claim, however, the link between the claim and the data they present needs to be articulated further. This is particularly important as this claim is the motivation for study 2.

8. In page 15 the authors decide to split the sample based on their responses to the dualism question. I have several methodological concerns about this analysis. First, it appears to me that this analysis is exploratory, and it would be useful if the authors labelled it as such. However, if this analysis is not exploratory, then it needs to be motivated in the hypotheses section. Second, the authors exclude participants whose attitude did not align with their experimental condition (e.g., they eliminated participants that rated below the midpoint only if they were in the dualist condition). Their reasoning for doing this was not explained. This decision is concerning because, as shown by the means in page 13, the averages for both conditions are below the midpoint. This creates an imbalance in the conditions where almost all the participants in the physicalist condition are kept in the analysis, but only one third of those in the dualist condition are kept in the analysis. Third, the methodological literature suggests that dichotomizing continuous variables often leads to lower power and higher spurious findings (see MacCallum, 2002). Rather than dichotomizing their predictors, the authors could examine the interaction between mind-body distance (as a continuous variable) and symptom condition in a regression.

9. In page 17, the authors claim that “our results suggest that (a) people are more likely to view visual symptoms of dyslexia as indicative of a hereditary brain disorder.” Although the hereditary portion is found in their data, the “brain disorder” claim was only significant in Study 3 (the p-values for studies 1 and 2 are above .100). There are other instances in the manuscript where results that do not reach the cut-off value of .050 are describe in the discussion in a way that implies they are significant. The authors should consider qualifying all such statements describing how they were not significant, or only significant in one study.

Minor issues

• Please report demographic information for the participants in the body of the text rather than in the supplemental materials. When I examined the supplemental materials, I found no information about participants’ age, gender and race or ethnic information. It would be useful to report this information. Additionally, the reporting of the demographic information in the supplemental materials combines all three studies (or at least that is my impression from reading it). Please report the demographic information separately for each study.

• Please include how much money participants received for participating in the studies and how long it took them to complete the studies.

• It is common to include attention checks in Mechanical Turk studies to make sure that participants are reading carefully. Did the authors include attention checks? Did all participants pass the checks?

• Where there any order effects?

• Please present the full range of the y-axis in Figures 2 and 3 (this would match the range shown in Figures 1, 4 and 5)

• If at all possible, it would be nice if the same color coding for the conditions is used in all the graphs.

• I would be useful to report the simple effects for the test by symptom interaction in the results section rather than in the discussion section (pg. 11).

• Has the diagram scale to measure mind-body distance been used before? How should this measure be interpreted? It looks like from 4 onwards people are selecting that the mind and body do not overlap, does it matter how separate the circles are?

• The authors describe in their introduction to part 2 that people are “inherent dualists,” however by looking at the means on their mind-body distance task, even the group primed to think in a dualistic manner had scores below the mid-point (suggesting they are more physicalist). How do the authors make sense of these conflicting findings?

• The footnote on page 15 should be incorporated into the body of the text.

• There are inconsistencies in the manuscript as to how many significant digits are reported for the p-values (ranges from 2 to 4).

• When conducting ANOVA’s please report whether they are between, repeated measures, or mixed.

• Please report standard deviation every time you report means.

• I ran the paper through statcheck.io and found three instances where the p-values were not consistent. Please double check you that all statistics are accurate.

• Why test each average against the mid-point of the scale? (I am not saying this is wrong, but rather that it needed some justification as it rarely came up in the results).

• There are stylistic and typographical issues in the manuscript (how quotations are cited, missing parentheses, extra punctuation).

Reviewer #2: This study had two main aims: (1) examining the relationship between peoples’ attitudes towards the symptoms of dyslexia (related to visual or phonological decoding difficulties) and their views on etiology and severity; (2) examining the relationship between the perceived mind-body linkage and the view of heritability of dyslexia with an implication of Dualism and Essentialism. To address these aims, the researchers designed 3 experiments. The 1st experiment studied the link between people’s attitudes towards the symptoms of dyslexia and its perceived etiology. Forty participants were asked to read two vignettes, featuring two individuals with reading difficulties associated with different symptoms, one with visual difficulties and another with phonological decoding. Participants were asked to evaluate the likelihood of reading disorder, severity and etiology. Eight t-tests were run to examine if the evaluation of reading disorder differed between individuals presented with visual-related vs. decoding-related symptoms. Overall the results showed that people tended to view visual difficulties (vs. decoding difficulties) as more indicative of reading disorder, more severe, more likely to be biologically caused, and more heritable. The 2nd experiment examined if people’s attitudes towards the etiology of reading disorder were influenced by whether the disorder was diagnosed by brain vs. behavioral measures. Forty participants read two vignettes: (1) one individual was diagnosed with dyslexia using brian test; (2) another individual was diagnosed with dyslexia using behavioral results. Participants were then asked seven questions on the etiology of reading disorder for each individual. Seven 2 Symptoms (Decoding/Visual) x 2 Test (Brain/Behavior) ANOVAS were run. Overall the results showed that when a diagnosis was done by brain test (vs. behavioral test), participants considered dyslexia as more innate, less likely to be caused by environment , more likely linked to biological basis, had poorer prognosis, more likely to affect the brain, and more likely to transfer to genetic clone. The 3rd experiment aimed at (1) replicating the results in Experiment 1, and (2) examining whether people’s attitudes towards the disorder would vary when the perceived mind-body linkage was different. A total of 120 participants were divided into 2 groups: Dualism group read an essay that describes the mind as distinct from the body; and Physicalism group read an essay that describes the mind and body as the same. Reading these essays served as a priming paradigm. Then, the participants read two vignettes, one featuring an individual suffering from a reading difficulty with decoding difficulty, and another featuring an individual suffering from visual symptoms. Similar to Experiment 1, participants were asked to evaluate the likelihood of a disorder, its severity, and etiology. Overall, the results showed people’s attitudes towards the severity of the disorder were modulated by the priming conditions. Participants who were primed with Dualism content considered reading disorders associated with visual symptoms as more severe than decoding difficulties. However, no significant difference in severity was observed when participants were primed in the Physicalism condition.

Overall, this study is interesting and the results will give us insight into the general public’s attitudes towards the etiology of dyslexia and the underlying causes of misconceptions about the disorder. The experiments, especially the vengitetes in the 3 experiments were well designed. Though the experiments are sound, the statistical analyses and the interpretation appear to be problematic. The researchers should address these fundamental issues; otherwise, the results and interpretation would not stand.

(1) The authors interpreted results with p-value = 0.5 as if it was significant. Furthermore, all the analyses had multiple t-tests and ANOVAS but no corrections were applied. If a multiple correction were to apply in the analyses, all results in Experiment 1 would be null.

(2) ANOVAS. The researchers did not mention what kind of anovas were used. Are they repeated measures? Clarification is needed.

6. PLOS authors have the option to publish the peer review history of their article (what does this mean?). If published, this will include your full peer review and any attached files.

Reviewer #1: **Yes: **David Menendez

Reviewer #2: No

---

## [Author Response · Author response to Decision Letter 0]

11 Jun 2021

Please find detailed reply in the attached file

---

## [Decision Letter · Decision Letter 1]

10 Sep 2021

PONE-D-20-39703R1Public misconceptions about dyslexia: the role of intuitive psychologyPLOS ONE

Dear Dr. Berent,

Thank you for submitting your manuscript to PLOS ONE. After careful consideration, we feel that it has merit but does not fully meet PLOS ONE’s publication criteria as it currently stands. Therefore, we invite you to submit a revised version of the manuscript that addresses the points raised during the review process.

Dear Pr. Berent,

I have now received the Reviewers' comments for your revised version. I concur with the Reviewers' comments that you did a great work on your Introduction section. I read it carefully and from my point of view, this is much clearer and theoretically-grounded. I would like to emphasize that your revised version addresses all the comments. And both Reviewers agreed.

However, both Reviewers would ask you to pay attention to some minor comments. They are not difficult to be addressed.

I have the feeling that you can fix them easily.

After this revision, I will not send your revised manuscript to the Reviewers and I will make my decision along with the Editor.

I sincerely think your manuscript will therefore be suitable for publication in PO.

I would like to congratulate you on the quality and, most significantly, on the originality of your paper.

You did a great work to respond to the Reviewers' comments.

I hope you will find these comments useful to submit a final version of your manuscript soon.

Regards,

N. Maïonchi-Pino

We look forward to receiving your revised manuscript.

Kind regards,

Norbert Maionchi-Pino, Ph.D

Academic Editor

PLOS ONE

Journal Requirements:

Additional Editor Comments (if provided):

Dear Pr. Berent,

I have now received the Reviewers' comments for your revised version. I concur with the Reviewers' comments that you did a great work on your Introduction section. I read it carefully and from my point of view, this is much clearer and theoretically-grounded. I would like to emphasize that your revised version addresses all the comments. And both Reviewers agreed.

However, both Reviewers would ask you to pay attention to some minor comments. They are not difficult to be addressed.

I have the feeling that you can fix them easily.

After this revision, I will not send your revised manuscript to the Reviewers and I will make my decision along with the Editor.

I sincerely think your manuscript will therefore be suitable for publication in PO.

I would like to congratulate you on the quality and, most significantly, on the originality of your paper.

You did a great work to respond to the Reviewers' comments.

I hope you will find these comments useful to submit a final version of your manuscript soon.

Regards,

N. Maïonchi-Pino

Reviewers' comments:

Reviewer's Responses to Questions

**Comments to the Author**

1. If the authors have adequately addressed your comments raised in a previous round of review and you feel that this manuscript is now acceptable for publication, you may indicate that here to bypass the “Comments to the Author” section, enter your conflict of interest statement in the “Confidential to Editor” section, and submit your "Accept" recommendation.

Reviewer #1: All comments have been addressed

Reviewer #2: (No Response)

2. Is the manuscript technically sound, and do the data support the conclusions?

Reviewer #1: Yes

Reviewer #2: Yes

3. Has the statistical analysis been performed appropriately and rigorously? 

Reviewer #1: Yes

Reviewer #2: No

4. Have the authors made all data underlying the findings in their manuscript fully available?

Reviewer #1: Yes

Reviewer #2: Yes

5. Is the manuscript presented in an intelligible fashion and written in standard English?

Reviewer #1: Yes

Reviewer #2: Yes

6. Review Comments to the Author

Reviewer #1: This is a very responsive revision. The authors have addressed all of the comments I made in my first review. The new introduction, although long, adequately situates the study and motives the hypotheses. The methods are now much clearer, and the results easier to read. All the statistics reported seem to be in order, so the authors did a good job at checking them.

I have only two minor comments.

1. The authors claimed in several sections that the people are more likely to “essentialize visual symptoms because they perceive them as more strongly anchored in the body” (page 21). Although this interpretation is supported by their findings, the authors never tested the mediation. Therefore, it might be beneficial to qualify some of these statements, as causality cannot be currently assessed.

2. Figures 1 and 2 did not display correctly on my screen. There was a weird ? box in several places which made it look weird and slightly challenging to read. I am assuming this was not intentional, so it might be worth looking into why it is happening.

Reviewer #2: This revision is a much improved version. The researchers have been very responsive with regard to reviewers' comments. The introduction is more theoretically grounded and coherent. My only question is that given the number of t-tests and ANOVAs being conducted, e.g., 7 repeated-measures ANOVAs, no multiple comparison correction was applied. What were the rationales? The p-values reported in the results may not stand the test of correction. I think it's fine given the exploratory nature of the study. It's better to include it, rather than reporting the "significant" results. Overall, I think this study is interesting, and i see the challenge of doing interdisciplinary research like this one. I think readers would be interested in learning more about how dualism and essentialism could be potentially related to public misconception of dyslexia.

7. PLOS authors have the option to publish the peer review history of their article (what does this mean?). If published, this will include your full peer review and any attached files.

Reviewer #1: **Yes: **David Menendez

Reviewer #2: No

---

## [Author Response · Author response to Decision Letter 1]

10 Sep 2021

Please find a detailed cover letter attached

---

## [Editor Report · Decision Letter 2]

12 Oct 2021

Public misconceptions about dyslexia: the role of intuitive psychology

PONE-D-20-39703R2

Dear Dr. Berent,

We’re pleased to inform you that your manuscript has been judged scientifically suitable for publication and will be formally accepted for publication once it meets all outstanding technical requirements.

Kind regards,

Norbert Maionchi-Pino, Ph.D

Academic Editor

PLOS ONE

Additional Editor Comments (optional):

Dear Pr. Berent,

First of all, I would like to thank you for replying to the comments addressed by the Reviewer.

You have satisfied the comments and I do not want to provide further advice or feedback that would delay publication of your manuscript.

As I had mentioned, I made my decision on your responses. And I am therefore very pleased to accept the manuscript for publication in PO.

Congratulations!

Thank you again for submitting your work to PO. I am looking forward to seeing it online shortly.

Best regards,

Norbert Maïonchi-Pino
---

## [Editor Report · Acceptance letter]

22 Nov 2021

PONE-D-20-39703R2 

Public misconceptions about dyslexia: the role of intuitive psychology 

Dear Dr. Berent:

I'm pleased to inform you that your manuscript has been deemed suitable for publication in PLOS ONE. Congratulations! Your manuscript is now with our production department. 

Kind regards, 

on behalf of

Dr. Norbert Maionchi-Pino 

Academic Editor

PLOS ONE